# Integrated Genomic Analysis Reveals the Impact of Avermectin on Chromatin Structure and Gene Expression Regulation in *Bombyx mori*

**DOI:** 10.3390/insects16030298

**Published:** 2025-03-12

**Authors:** Yongkang Guo, Tong Zhang, Wei Lu, Dan Liu, Junjie Lao, Na Zhang, Hao Sun, Ling Jia, Sanyuan Ma

**Affiliations:** Integrative Science Center of Germplasm Creation in Western China (CHONGQING) Science City, Biological Science Research Center, Southwest University, Chongqing 400715, China; gyk666@email.swu.edu.cn (Y.G.); zt137703197@email.swu.edu.cn (T.Z.); luw@office365.swu.edu.cn (W.L.); ld1957252393@163.com (D.L.); laojunjie2020@163.com (J.L.); zhangna1522@outlook.com (N.Z.); sunhao19971204@163.com (H.S.)

**Keywords:** Avermectin, ATAC-seq, *Bombyx mori*, chromatin structure, gene expression, Hi-C, RNA-seq

## Abstract

This study examines the effects of avermectin, a widely utilized insecticide, on silkworm (*Bombyx mori*), with a specific focus on its influence on genomic architecture and gene expression. As a common pesticide employed for pest control, avermectin, while effective, raises ecological concerns regarding its potential impact on non-target organisms, including silkworms. In this investigation, we conducted a comparative analysis between silkworms fed with untreated mulberry leaves (control group) and those exposed to avermectin-treated leaves (experimental group). Tissue samples were collected from the silkworms’ digestive systems following pesticide exposure. Through a comparative analysis of DNA organization patterns and gene activity between the experimental and control groups, we observed significant alterations in chromatin structure. Notably, silkworms exposed to avermectin exhibited a more relaxed chromatin configuration, accompanied by intensified interactions between short-range DNA elements. Furthermore, we identified pronounced changes in the 3D spatial organization of DNA within detoxification-related genes, which were associated with elevated gene expression levels. These findings contribute valuable insights into the mechanisms underlying pest resistance to pesticides and lay a foundation for future research into the adaptive responses of insects to environmental stressors.

## 1. Introduction

Avermectin is a class of macrocyclic lactones that is derived from *Streptomyces avemitilis* and is widely used in agriculture owing to its potent insecticidal and anthelmintic activities [1]. This compound exerts its effects by binding to the glutamate-gated chloride channels (GluCls) in invertebrate nerves and muscle cells, leading to hyperpolarization, paralysis, and eventual death of the target pests [1,2]. Avermectin has become a crucial component of integrated pest management (IPM) strategies because of its great efficacy and minimal toxicity to mammals, which is linked to its failure to pass the blood–brain barrier [1]. However, the extensive application of avermectin-based insecticides has led to the development of resistance in various arthropod pests, such as *Plutella xylostella* [3,4,5,6] and *Tetranychus urticae* [7,8,9], among others, often mediated by cytochrome P450-mediated detoxification or mutations in target-site genes [2,10,11]. Although the mechanisms of resistance in target pests have been partially elucidated, the possible genomic and epigenetic impacts of avermectin exposure on non-target species, particularly beneficial insects such as silkworms, remain poorly understood.

The silkworm (*Bombyx mori*) is a lepidopteran model organism with significant economic and scientific value and can be used to investigate these impacts [12,13]. As a close relative of agricultural pests, silkworms share conserved physiological and genomic features with other lepidopterans, making it an ideal proxy for studying insecticide interactions [13,14]. The silkworm midgut, a critical site for nutrient absorption, detoxification, and immune responses, is directly exposed to dietary toxins, including pesticides [15,16]. Previous studies have demonstrated that pesticide exposure disrupts midgut physiology; for example, acetamiprid induces oxidative stress, inflammation, and the deregulation of metabolic pathways in the silkworm [15]. However, the effect of pesticide-induced stress on the three-dimensional (3D) genomic architecture has not been investigated, despite being a key regulator of gene expression and cellular function.

Recent advances in 3D genomics such as high-throughput chromosome conformation capture (Hi-C) [17], RNA sequencing (RNA-seq) [18], and Assay for Transposase-Accessible Chromatin using sequencing (ATAC-seq) [19] have revolutionized our understanding of how spatial genome organization influences gene regulation during development, differentiation, and environmental challenges [20,21]. These technologies have been instrumental in analyzing host–pathogen interactions in silkworms, including the role of super-enhancers in *Bombyx mori* nucleopolyhedrovirus (BmNPV) infection [21]. Building on these approaches, our study aimed to integrate Hi-C, ATAC-seq, and RNA-seq to investigate avermectin-induced 3D genome remodeling in the silkworm midgut.

Previous studies have indicated that the LC50 value of avermectin for 4-day-old larvae is 16.0 μg/L after 96 h exposure [22]. A half-LC50 dose is commonly used as a sublethal concentration because it balances minimal mortality with significant physiological stress, enabling the study of early molecular and phenotypic responses without the confounding effects of high mortality. Thus, in this study, we exposed the larvae of silkworms to sublethal doses (8 μg/L) of avermectin and performed multi-omics analyses of the midgut tissues. Multi-omics analysis revealed significant reorganization of chromatin compartments, disruption of topologically associating domains (TADs), and extensive changes in chromatin conformation. These alterations were accompanied by an increase in chromatin accessibility, enhanced spatial interactions, and widespread upregulation of gene expression, particularly in regions carrying the detoxification-related ATP-binding cassette (ABC) transporter genes. These findings reveal a previously unrecognized link between 3D genome dynamics and pesticide-induced stress responses in insects. This study not only advances our understanding of the silkworm’s response to avermectin but also provides a framework for evaluating the genomic impacts of agrochemicals on non-target species for sustainable pest management.

## 2. Materials and Methods

### 2.1. Bombyx mori Feeding and Pesticide Application

The silkworm strains used in this study, specifically the Dazao strain, were provided by the Biological Science Research Center of Southwest University. All the samples were fed mulberry leaves sourced from the mulberry garden of the university. Both the control and experimental groups were raised under the same conditions at 25 °C. In the experimental group, mulberry leaves were soaked in an 8 μg/L avermectin working solution for 20 min and then air-dried. The working solution was prepared by diluting a 6% avermectin stock solution (0.6 g of 97% pure avermectin dissolved in 10 mL DMSO) with 0.1% ddH_2_O, resulting in a final concentration of 8 μg/L. From the first day of the fifth instar (L5D1), silkworms were fed these treated mulberry leaves, which were cut into pieces and provided three times daily (morning, noon, and evening). After 24 h of exposure, midgut tissues were collected, ensuring the removal of the peritrophic membrane [22].

### 2.2. Generation of Hi-C Libraries and Analysis of Sequencing Data

A Hi-C library was generated from the L5D2 midgut tissues of silkworms. The samples were fixed with formaldehyde to cross-link chromatin, followed by fragmentation using the MboI restriction enzyme. Biotinylated nucleotides were incorporated during the ligation process to enable subsequent capture of the ligation products. The libraries were sequenced using a high-throughput platform (Chongqing Allbioknow Biotechnology Co., Ltd., Chongqing, China) to generate paired-end reads. The Hi-C data were processed using the snHiC pipeline, which integrates several software tools for efficient analysis [23]. These reads were aligned to the silkworm genome assembly (Silkbase) [24], and only chromosomes 1–28 were used, excluding the scaffold. The parameters *matrix_resolution* and *restriction_enzyme* were set to *[1,10,20,40,50,100]* and *MboI*; the parameters *TAD_caller* and *loop_caller* were set to *HiCexplorer*; the parameters *perform_grouped_analyses*, *detect_components*, and *perform_differential_contacts_analyses* were set as *true*. The details of snHiC config can be found in https://github.com/luxiangze/Avermectin-Bombyx_mori_Analysis/blob/main/hic_snHiC/snHiC_config.yaml (Accessed on 10 March 2025).

### 2.3. RNA Sequencing and Data Analysis

RNA sequencing libraries were prepared using total RNA extracted from six midgut samples (three each from the control and experimental groups) using the TRIzol method, followed by purification using an RNA clean concentrator kit. The libraries were subjected to quality control and sequenced to generate 150 bp paired-end reads on the same high-throughput platform. The raw sequencing data were processed using the nf-core/rnaseq (v3.18.0) pipeline, which is used to analyze RNA sequencing data obtained from organisms with a reference genome and annotation. The parameters *aligner*, *save_reference*, and *skip_pseudo_alignment* were set to *star_salmon*, *true*, and *true*, respectively. The details of the nf-core/rnaseq pipeline parameters can be found in https://github.com/luxiangze/Avermectin-Bombyx_mori_Analysis/blob/main/rna_nf/nf-params.json (Accessed on 10 March 2025).

Differential gene expression analysis was performed using the differential abundance (v1.5.0) pipeline of nf-core, including the DESeq2 package [25], with a significance threshold set at an adjusted *p*-value (*p*-adj) < 0.05 and an absolute log2 fold change cutoff of 1. Gene Ontology (GO) enrichment and Kyoto Encyclopedia of Genes and Genomes (KEGG) pathway analyses of the differentially expressed genes were conducted using the cluster Profiler (v4.14.4) R package [26].

### 2.4. Processing and Visualization of ATAC-seq Data

ATAC-seq libraries were generated from the same midgut samples using an adapted version of the original protocol [27]. Briefly, midgut tissues were ground in liquid nitrogen or homogenized, and cells were resuspended in pre-chilled PBS, washed, and filtered through a 40 μm cell strainer. Nuclei were extracted using a lysis buffer, washed, and resuspended in the Tn5 transposition reaction mix. The transposition was carried out at 56 °C for 30 min. Following transposition, DNA fragments were recovered, amplified by PCR, purified, and assessed for library fragment size distribution. Similarly, sequencing of the ATAC-seq libraries was performed, and the resulting data were processed using the nf-core/atacseq (v2.1.2) to identify the peaks of chromatin accessibility, and the parameters *macs_gsize*, *macs_fdr*, and *skip_picard_metrics* were set to *460000000*, *0.05*, and *true*, respectively. The details of nf-core/atacseq pipeline parameters can be found in https://github.com/luxiangze/Avermectin-Bombyx_mori_Analysis/blob/main/atac_nf/nf-params.json (Accessed on 10 March 2025).

## 3. Results

### 3.1. Construction of a 3D Genomic Interaction Map for Bombyx mori

The 3D structure of the genome of the silkworm is integral to its gene regulation mechanisms. Compartmentalization into active and inactive regions, coupled with loop extrusion dynamics and epigenetic modifications, creates a complex regulatory landscape that facilitates the precise control of gene expression during development and adaptation [21,28]. High-throughput chromosome conformation capture techniques (Hi-C) allow for the genome-wide capture of chromatin interactions, providing a detailed view of how different regions of DNA come into contact within the nucleus [29]. Assay for Transposase Accessible Chromatin using sequencing (ATAC-seq) is a crucial technique for studying chromatin accessibility; this method allows researchers to identify regions of the genome that are accessible for transcription factor binding, thereby influencing gene transcriptional activity.

To investigate the impact of the 3D genome structure of silkworms on gene expression, we constructed a 3D interaction map using Hi-C, ATAC-seq, and RNA-seq (Figure 1A,B). A total of 43,482 highly accessible chromatin regions (peaks) were identified. Notably, the regions with high chromatin accessibility were predominantly concentrated around the transcription start sites (TSSs) (Figure 1C,D). Although promoters accounted for only 2.75% of the silkworm genome under normal conditions [21], they constituted 30.87% of the identified highly accessible chromatin regions (Figure 1E).

In terms of 3D genomic interactions, we identified 1028 topologically associating domains (TADs) at a resolution of 20 kb and 981 loops at a resolution of 10 kb. Among the 981 high-frequency interaction regions, 1337 associated genes were identified and subjected to functional analyses (Figure 1F,G). The results demonstrate that the 3D genome architecture of the larval midgut of the silkworm plays a significant regulatory role in neural system development, cytoskeletal dynamics, and signal transduction. These functions are closely associated with larval growth, development, and environmental adaptation.

### 3.2. Avermectin Application Causes Chromatin Conformational Changes in Silkworm

Avermectin is a class of antiparasitic agents derived from the soil bacterium *S. avermitilis*, and reports have shown that it causes high mortality in silkworm larvae [22]. Based on the 96 h median lethal dose (LC50) of 16 μg/L avermectin, an 8 μg/L concentration was selected for this experiment. After three days of feeding on the treated leaves, the silkworms exhibited severe toxic symptoms, including anorexia, salivation, body twisting into C or S shapes, and eventual immobility leading to death. In contrast, the control group, fed with untreated leaves, showed normal growth and development. These results indicate that even low concentrations of avermectin cause significant toxicity in silkworms. To determine how the silkworm genome regulates gene expression in response to avermectin stress, we constructed 3D genome differential interaction maps for the experimental and control groups (Figure 2A). Results showed that the 3D genome architecture of the experimental group was more compact, with more frequent short-range interactions, compared to that of the control group (Appendix A).

We observed that the experimental group contained approximately 20% more loops than the control group when we performed loop detection on the experimental and control groups separately (Appendix A). After excluding loops that overlapped at identical genomic positions, 943 differential loops were identified. Moreover, the distribution of loops between the two sample groups after avermectin treatment revealed that the loops in the silkworm genome became more compact and denser. Randomly selected DNA interaction maps suggest that gene regulation occurs more frequently and intensively in silkworms. In these maps, deeper red staining indicates higher interaction intensity. (Figure 2B–E).

To further analyze the gene expression differences in the experimental group, we conducted a differential gene expression analysis between the two sample groups. The results showed that the number of up-regulated genes was twice that of the down-regulated genes in the differentially expressed gene set (Figure 2F, Appendix A). Functional analysis of the differentially expressed genes (Figure 2F) revealed that avermectin treatment might affect neural regulation, energy metabolism, ribosome biogenesis, nucleocytoplasmic transport, and mitophagy in silkworms, thereby leading to cellular stress responses and metabolic disorders (Figure 2G,H). These findings indicate that after avermectin treatment, the intensity of spatial interactions in silkworms increases to enhance the expression of functional genes related to metabolism.

### 3.3. Silkworm Resistance to Avermectin Is Regulated by Chromatin Conformation

To elucidate the detailed regulatory mechanisms of genome interactions in silkworms, we performed ATAC-seq differential analysis between the control and experimental groups. The results showed increased chromatin accessibility near the transcription start sites (TSSs) in the experimental group, but there were no significant changes in the number of genes (Figure 3A–C). Typically, chromatin-accessible regions (peaks) near TSSs can be considered to be potential transcription factor-binding sites. Analysis of the peaks near the TSSs showed a decrease in the proportion of long-range peaks (>10 kb) and an increase in the proportion of short-range peaks (0–5 kb). This finding is consistent with our previous Hi-C data and indicates an increase in proximal interactions and regulation (Figure 3D).

We further analyzed the genomic distribution characteristics of the peaks and found that short-range peaks increased, whereas the proportion of peaks located in the promoter regions decreased. Conversely, the proportion of peaks located in the intergenic regions increased. Hence, we speculate that these proximal intergenic regions may be associated with enhancers (Figure 3E).

To further analyze the differences in chromatin accessibility between the two groups, we generated an UpSet plot to visualize the differences in the shared peaks between the two samples (Figure 3F). The results show that the two sample groups share a total of 43,982 peaks, among which 4319 peaks are present in every sample. Additionally, 90 peaks are exclusively detected in the control group, while 446 peaks are unique to the experimental group. To gain a deeper insight into the peaks that increased in the experimental group, we identified 446 peaks that were present exclusively in the experimental group but absent in the control group (Figure 3G). These peaks were predominantly located in the intergenic regions and were mostly close to the TSS (Figure 3H). Motif analysis identified 16 significantly enriched motifs (*E*-value < 0.05), potentially representing binding sites of key transcription factors. Among the 435 genes associated with these 446 accessible chromatin peaks, three genes (KWMTBOMO00840, KWMTBOMO07716, KWMTBOMO10231) showed significantly up-regulated expression. Notably, *MTX2*(KWMTBOMO00840) is likely involved in the detoxification metabolism of the silkworm. Overall, our results suggest that silkworms respond to avermectin exposure by increasing chromatin accessibility, particularly in potential enhancer regions near the TSS, thereby enhancing the expression of genes related to functional adaptation.

### 3.4. Three-Dimensional Structural Rearrangement Plays an Important Role in Gene Expression in Silkworm

To further investigate the role of 3D genome structural changes in the regulation of gene expression in silkworms, we conducted a genome-wide differential spatial interaction analysis. The number of differential spatial interactions varied across different resolutions, with 9573 differential interaction pairs identified at a 1 kb resolution and 814,991 at a 10 kb resolution (Appendix A).

At 10 kb resolution, a total of 13,663 genes were involved in differential spatial interactions. Among them, 22 homologous genes previously reported to be associated with avermectin resistance in other species were identified. Notably, three of these genes exhibited significant differential expression and were up-regulated in the transcriptome analysis (Appendix A), including two genes related to ABC transporters (Figure 4A,B). Additionally, among the genes involved in differential spatial interactions, 199 were up-regulated (Appendix A, Appendix A).

Among the 199 up-regulated genes, we identified two genes (KWMTBOMO00492: *LPCAT3*, KWMTBOMO07140: *AGO2*) that not only exhibited differential spatial interactions but also underwent changes at the topologically associating domain (TAD) level (Figure 4C,D). Functionally, these two genes are involved in detoxification and stress resistance mechanisms in silkworms. In conclusion, the three-dimensional structural changes in the silkworm genome play an indispensable role in gene regulation.

## 4. Discussion

An understanding of the impact of external stressors such as avermectin on the 3D genome architecture of silkworms provides fundamental insights into the molecular dynamics governing gene expression and insect physiology. Our findings revealed an interplay between chromatin conformation, gene regulation, and organismal responses to pesticide exposure.

The increased chromatin accessibility and consequent reconfiguration of 3D interactions suggest a robust adaptive response that enables silkworms to enhance their expression of genes that are crucial to their survival under stress conditions. The predominance of the up-regulated genes associated with metabolic processes, cellular stress responses, and detoxification pathways underscores the coordinated genomic response aimed at mitigating the effects of avermectin toxicity.

Previous studies have shown that chromatin structure is not static; it undergoes dynamic changes in response to environmental stimuli [30]. Avermectin exposure appears to stimulate a transition towards a more compact chromatin architecture to enhance short-range interactions, particularly near gene promoter regions. This shift likely facilitates the recruitment of transcription factors and assembly of the transcriptional machinery to effectively amplify the gene expression of numerous adaptive genes. The identification of novel enhancer regions in our ATAC-seq analysis further supports the theory that silkworms strategically utilize enhancers to fine-tune gene expression in response to environmental challenges.

Notably, the chromosomal rearrangements observed in this study may explain the variations in sensitivity to avermectin among different insects [31]. The differential expression of ABC transporter genes, which are often implicated in insecticide resistance [32], demonstrates the potential mechanism by which these proteins confer a survival advantage to populations subjected to avermectin application. Hence, there is a need to explore the genetic diversity and functional genomics of silkworm populations, because varying responses to insecticides may result from subtle differences in 3D genome organization and chromatin remodeling dynamics.

Our study emphasizes the importance of using integrative genomic approaches to elucidate the complex behaviors of the genome under stress. By combining Hi-C, RNA-seq, and ATAC-seq data, we developed a comprehensive picture of the alterations in chromatin structure and gene expression induced by avermectin treatment. These changes in spatial genomic organization are not merely incidental but are pivotal in orchestrating a targeted genomic response that enhances the ability of the silkworm to cope with pesticide exposure.

## 5. Conclusions

Our study shows how avermectin affects the 3D genome structure and gene expression in the silkworm midgut. Using Hi-C, ATAC-seq, and RNA-seq, we found changes in chromatin organization, increased accessibility, and shifts in genes related to detoxification and stress response. This study reveals a clear link between pesticide stress and genome architecture in silkworms. Future research could explore how other pesticides impact different non-target organisms. Comparing responses across species will help identify common and unique stress mechanisms. Studying long-term and inherited effects of pesticide exposure on genome structure is also important.

## Figures and Tables

**Figure 1 insects-16-00298-f001:**
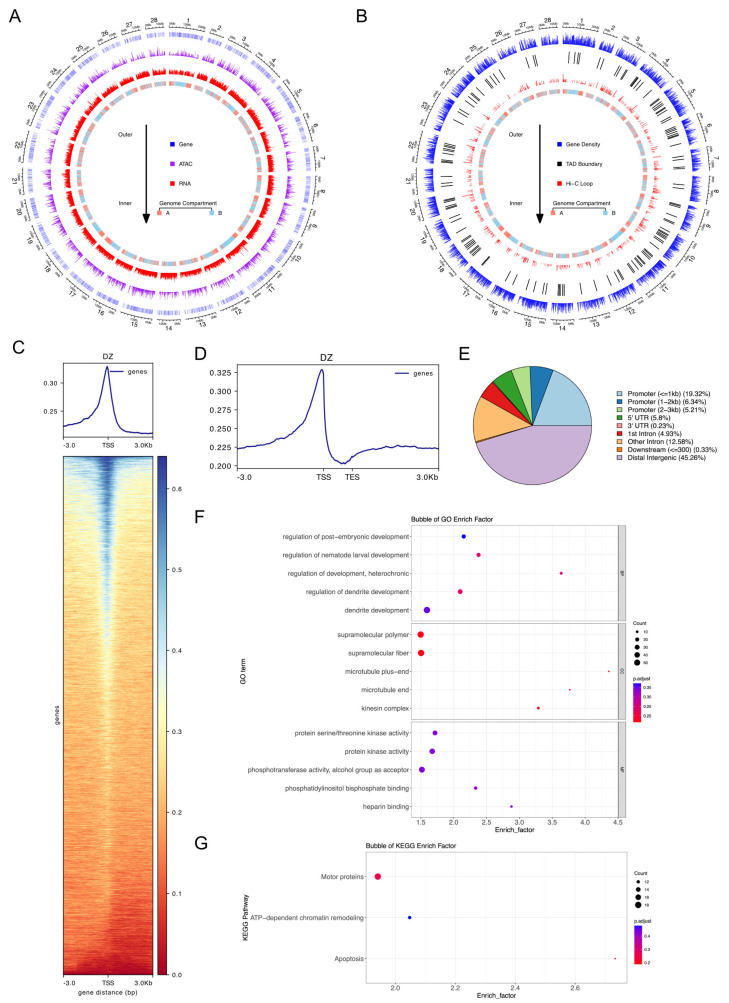
The genome structure of the silkworm is closely related to gene expression. (**A**) The accessibility and expression map of the silkworm genome. (**B**) The interaction map of the silkworm genome. (**C**) Chromatin accessibility around all 16,880 transcription start sites (TSSs) in the silkworm genome. (**D**) Chromatin accessibility within ±3 kb of the transcription start sites (TSSs) and transcription end sites (TESs) of all genes in the silkworm genome. (**E**) Distribution of chromatin accessibility peaks across the silkworm genome. (**F**,**G**) GO enrichment (**F**) and KEGG pathway (**G**) analysis of genes associated with chromatin accessibility peaks in the silkworm.

**Figure 2 insects-16-00298-f002:**
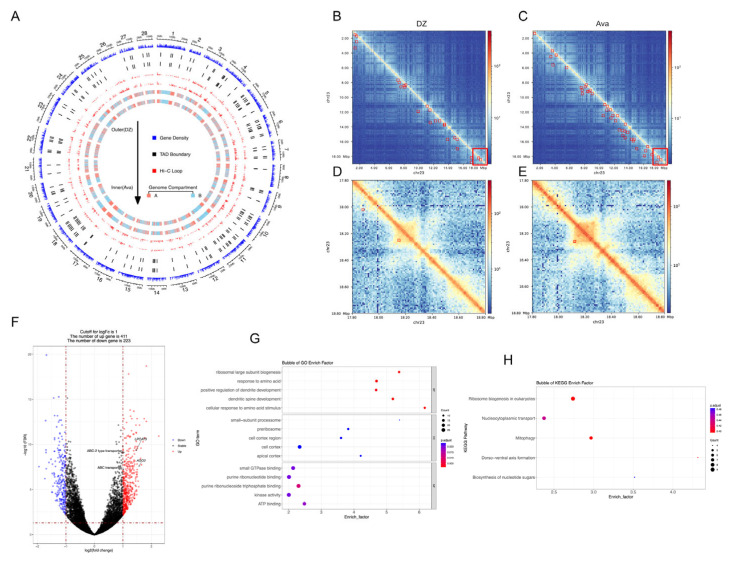
Genome-wide interaction differences in the silkworm lead to differential gene expression. (**A**) Genome interaction map between the experimental group (Ava) and the control group (DZ). (**B**–**E**) Differential loops between the experimental group and the control group in the silkworm. (**F**) Volcano plot of differentially expressed genes in the silkworm, with *p*_adj_ < 0.05. (**G**,**H**) GO enrichment (**G**) and KEGG pathway (**H**) analysis of differentially expressed genes in the silkworm.

**Figure 3 insects-16-00298-f003:**
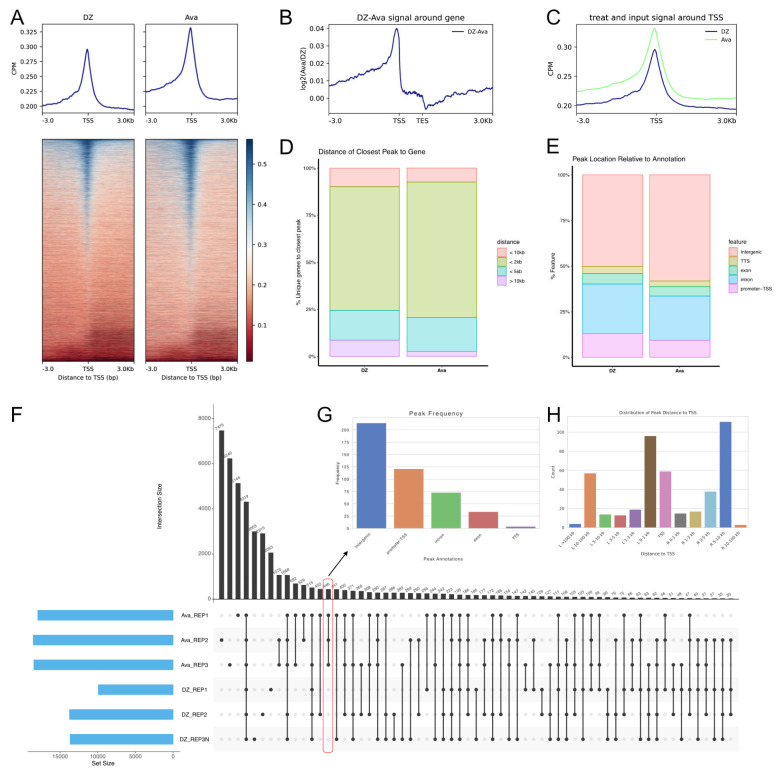
Changes in genome accessibility of the silkworm following avermectin treatment. (**A**–**C**) Differences in chromatin accessibility within 3 kb of the transcription start sites (TSSs) of all 16,880 genes between the control group (Ava) and the experimental group (DZ) (**A**,**C**), as well as differences in chromatin accessibility within 3 kb of both TSS and transcription end sites (TESs). (**B**,**D**) Distribution of chromatin accessibility peaks relative to their associated genes in the control group (Ava) and experimental group (DZ), categorized by distance: <10 kb, <5 kb, <2 kb, and >10 kb. (**E**) Differences in the positional distribution of chromatin accessibility peaks between the control group (Ava) and the experimental group (DZ). (**F**) Upset plot of chromatin accessibility peaks. The index of the vertical axis of the graph “Intersection Size” means the number of peaks. (**G**,**H**) Genomic distribution frequency histogram of chromatin accessibility peaks unique to the experimental group (Ava) (**G**), and distance distribution histogram of chromatin accessibility peaks from TSSs (**H**).

**Figure 4 insects-16-00298-f004:**
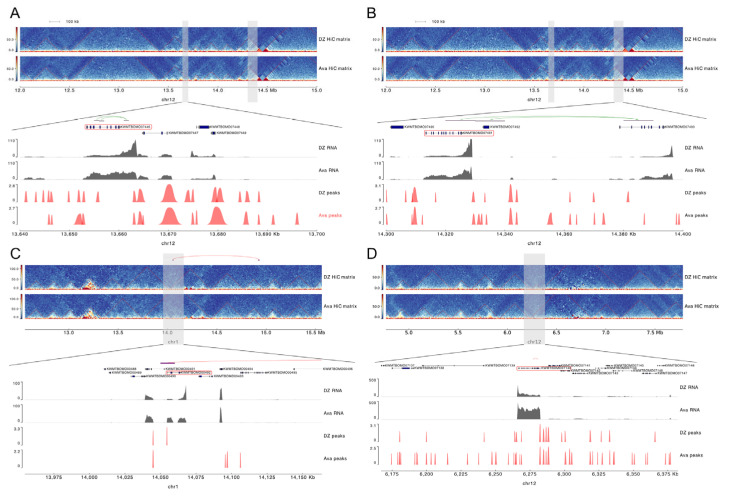
Three-dimensional conformation changes in the silkworm genome lead to up-regulation of detoxification-related genes. Interactions of detoxification genes ABC transporter proteins (**A**,**B**), *LPCAT3* (**C**), and *AGO2* (**D**) on the genome, transcript expression levels, and normalized epigenetic feature profiles.

## Data Availability

The images shown in the article were combined and modified using Affinity Designer2 (v2.5.7). The codes and scripts used in this study are all located in the GitHub repository. All the raw sequencing data generated in this study have been deposited in the NCBI BioProject database under the accession number PRJNA1225773.

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
