# Peer review of "Integrated Genomic Analysis Reveals the Impact of Avermectin on Chromatin Structure and Gene Expression Regulation in Bombyx mori"

_insects, 2025, doi:10.3390/insects16030298_

Round 1

Reviewer 1 Report

Comments and Suggestions for Authors

This study combines Hi-C, RNA-seq, and ATAC-seq technologies in a new way. It systematically explores how avermectin exposure affects the 3D genome structure and transcriptional regulation in the midgut of Bombyx mori. The research is well-organized and based on strong datasets. It gives useful knowledge for insect toxicology and pesticide risk assessment. Although the conclusions are convincing, some parts need to be made clearer or expanded to make the science more rigorous. Here are some specific suggestions.

  1. Please clarify if the concentration of avermectin used is a sublethal dose and state how long the exposure lasted. Also, briefly explain why this dose was chosen. This will help us better understand the effects that were observed.
  2. Provide a more comprehensive description of the Hi-C data analysis pipeline , for example adding the parameters for snHiC and ATAC-seq experimental conditions like the Tn5 transposase treatment duration/fragmentation to ensure reproducibility.
  3. Although the upregulation of ABC transporters is related to detoxification, it's important to stress the clear connections between the changes in chromatin accessibility and the differences in gene expression. Pointing out the conserved regulatory motifs in the accessible areas, such as transcription factor binding sites, will make this part of the study more robust.
  4. For figures modifying:

  - Label key DEGs in volcano plots in Figure 2 

  - Clarify proportions of shared/unique peaks in UpSet plots in Figure 3.

  - Ensure that all figures are large and clear enough to be easily read.

Addressing the above points will enhance clarity and broaden the impact of this important study. I recommend acceptance after minor revisions.

Author Response

Comments 1: Please clarify if the concentration of avermectin used is a sublethal dose and state how long the exposure lasted. Also, briefly explain why this dose was chosen. This will help us better understand the effects that were observed.
Response 1: Thank you for pointing this out. We agree with this comment. Therefore, we have added detailed information on the sublethal dose and provided a brief explanation of why this dose was selected. Additionally, we have supplemented a detailed description of the treatment conditions for individual silkworms. The specific revisions are as follows:
We have added “Previous studies have indicated that the LC50 value of avermectin for 4-day-old larvae is 16.0 μg/L after a 96-hour exposure. A half-LC50 dose is commonly used as a sublethal concentration because it balances minimal mortality with significant physiological stress, enabling the study of early molecular and phenotypic responses without the confounding effects of high mortality.” to briefly explain why this dose was chosen on page 2, lines 81-85. The related reference was also added in the revised manuscript on page 12, lines 411-413.
And we have changed “In the experimental group, mulberry leaves were soaked in avermectin solution for 12 h. Midgut tissue samples were collected from the silkworm larvae during the fifth instar, three days.” to “In the experimental group, mulberry leaves were soaked in an 8 μg/L avermectin working solution for 20 minutes and then air-dried. The working solution was prepared by diluting a 6% avermectin stock solution (0.6 g of 97% pure avermectin dissolved in 10 mL DMSO) with 0.1% ddH2O, resulting in a final concentration of 8 μg/L. From the first day of the fifth instar (L5D1), silkworms were fed these treated mulberry leaves, which were cut into pieces and provided three times daily. After 24 hours of exposure, midgut tissues were collected, ensuring the removal of the peritrophic membrane.” to explain in detail the processing conditions and procedures of the sample on page 3, lines 102-110.

Comments 2: Provide a more comprehensive description of the Hi-C data analysis pipeline , for example adding the parameters for snHiC and ATAC-seq experimental conditions like the Tn5 transposase treatment duration/fragmentation to ensure reproducibility.
Response 2: Thank you for pointing this out. We agree with this comment. Therefore, We have added more detailed parameter descriptions for the snHiC pipeline in the Methods section and specified the location of all relevant parameters in the associated GitHub repository. Additionally, we have provided a more comprehensive description of the ATAC-seq library construction method. The specific revisions are as follows:
We have changed “The TAD caller and loop detection used HiCexplorer, with parameters such as perform_grouped_analyses, detect_components, and perform_differential_contacts_analyses set as true.” to “The parameters matrix_resolution and restriction_enzyme were set to [1, 10, 20, 40, 50, 100] and MboI; the parameters TAD_caller and loop_caller were set to HiCexplorer; the parameters perform_grouped_analyses, detect_components, and perform_differential_contacts_analyses were set as true. The detail of snHiC config can find in snHiC_config.yaml” in page 3 line 120-124.
Then we supplemented the content of ATAC-seq sample processing in the text, revising "Briefly, the nuclei were extracted and subjected to Tn5 transposition, followed by amplification, and then purification." to "Briefly, midgut tissues were ground in liquid nitrogen or homogenized, and cells were resuspended in pre-chilled PBS, washed, and filtered through a 40 μm cell strainer. Nuclei were extracted using a lysis buffer, washed, and resuspended in the Tn5 transposition reaction mix. The transposition was carried out at 56°C for 30 mins. Following transposition, DNA fragments were recovered, amplified by PCR, purified, and assessed for library fragment size distribution." on page 4, lines 143-148. In addition, we have added “The detail of nf-core/rnaseq pipeline parameters can be found in rna_nf/nf-params.json” and “The detail of nf-core/atacseq pipeline parameters can be found in atac_nf/nf-params.json” on page 3, lines 133-134, and page 4, lines 152-153.

Comments 3: Although the upregulation of ABC transporters is related to detoxification, it's important to stress the clear connections between the changes in chromatin accessibility and the differences in gene expression. Pointing out the conserved regulatory motifs in the accessible areas, such as transcription factor binding sites, will make this part of the study more robust.
Response 3: Agree. We have, accordingly, performed motif analysis on chromatin-accessible regions specific to the experimental group, identifying 16 distinct motifs and providing the differentially expressed genes associated with the differential peaks. The specific revisions are as follows:
We have added “Motif analysis identified 16 significantly enriched motifs (E-value < 0.05), potentially representing binding sites of key transcription factors. Among the 435 genes associated with these 446 accessible chromatin peaks, three genes (KWMTBOMO00840, KWMTBOMO07716, KWMTBOMO10231) showed significantly up-regulated expression. Notably, MTX2 (KWMTBOMO00840) is likely involved in the detoxification metabolism of the silkworm.” to emphasize this point on page 7, lines 253-259.

Comments 4: For figures modifying:
- Label key DEGs in volcano plots in Figure 2 
- Clarify proportions of shared/unique peaks in UpSet plots in Figure 3.
- Ensure that all figures are large and clear enough to be easily read.
Response 4: Thank you for pointing this out. We agree with this comment. Therefore, we have added label key DEGs in volcano plots of Figure 2. We also have revised the figures size to make it easy to read and added proportions of shared/unique peaks of the UpSet plots in Figure 3. The specific revisions are as follows:
We add “The results show that the two sample groups share a total of 43,982 peaks, among which 4,319 peaks are present in every sample. Additionally, 90 peaks are exclusively detected in the control group, while 446 peaks are unique to the experimental group.” on page 7, lines 247-250.

Reviewer 2 Report

Comments and Suggestions for Authors

This MS entitled “Integrated Genomic Analysis Reveals the Impact of Avermectin on Chromatin Structure and Gene Expression Regulation in Bombyx mori” by Guo and colleagues evaluated the epigenetic effects of avermectin treatment. Genomic DNA conformation and mRNA transcription were analysed by high-throughput sequencing. Avermectin treatment induced a conformational change in the DNA, which could induce gene expression of some detoxification genes. It seems to cause resistance to avermectin. This MS provides much HT seq data and is consistent.

In this MS, no major problems were found in the experimental or static methods. Some minor points are listed below,

  1. All figures are too small to see. Please make lager.
  2. In line 112-115, the sentence about the significance threshold is doubled, please correct it.
  3. In results section, please add a description of the condition of the individual treated with avermictin.
  4. Figures 2B and C show the data from chr 23. Please state in the MS why the data from chr 23 were chosen.
  5. "Intersection Size" is the index of the vertical axis of the graph. What does it mean? Does it mean the number of peaks?
  6. In line 243-244. A reference is required.

Author Response

Comments 1: All figures are too small to see. Please make lager.
Response 1: Agree. We have, accordingly, revised the figures size to make it easy to read.

Comments 2: In line 112-115, the sentence about the significance threshold is doubled, please correct it.
Response 2: Thank you for pointing this out. We agree with this comment and have deleted the doubled sentence in the revised manuscript. Please see pages 3-4, lines 135-138.

Comments 3: In results section, please add a description of the condition of the individual treated with avermictin.
Response 3: Thank you for pointing this out. We agree with this comment. Therefore, we have added detailed information on the avermectin treatment conditions for individual silkworms. The specific revisions are as follows:
We have changed “In the experimental group, mulberry leaves were soaked in avermectin solution for 12 h. Midgut tissue samples were collected from the silkworm larvae during the fifth instar, three days.” to “In the experimental group, mulberry leaves were soaked in an 8 μg/L avermectin working solution for 20 minutes and then air-dried. The working solution was prepared by diluting a 6% avermectin stock solution (0.6 g of 97% pure avermectin dissolved in 10 mL DMSO) with 0.1% ddH2O, resulting in a final concentration of 8 μg/L. From the first day of the fifth instar (L5D1), silkworms were fed these treated mulberry leaves, which were cut into pieces and provided three times daily (morning, noon, and evening). After 24 hours of exposure, midgut tissues were collected, ensuring the removal of the peritrophic membrane.” on page 3, lines 103-110.

Comments 4: Figures 2B and C show the data from chr 23. Please state in the MS why the data from chr 23 were chosen.
Response 4: Thank you for pointing this out. We agree with this comment. Therefore, we have revised the description of the results for Figures 2B and 2C. The specific revisions are as follows:
We have changed “This suggests that gene regulation occurred more frequently and intensively in silkworms.” to “Randomly selected DNA interaction maps suggest that gene regulation occurs more frequently and intensively in silkworms. In these maps, deeper red staining indicates higher interaction intensity.” on page 6, lines 209-211.

Comments 5: "Intersection Size" is the index of the vertical axis of the graph. What does it mean? Does it mean the number of peaks?
Response 5: Yes, the “Intersection Size” represents the number of peaks. Thank you for pointing this out. We have added the statement “The index of the vertical axis of the graph “Intersection Size” represents the number of peaks” in the caption of Figure 3F to clarify the meaning of the index on the vertical axis on page 8, lines 270-271.

Comments 6: In line 243-244. A reference is required.
Response 6: Thank you for pointing this out. We agree with this comment. Therefore, we have added a reference on page 10, line 320-321.

Reviewer 3 Report

Comments and Suggestions for Authors

The paper entitled "Integrated Genomic Analysis Reveals the Impact of Avermectin on Chromatin Structure and Gene Expression Regulation in Bombyx mori" addresses an important topic that highlights the importance of Bombyx mori as a model organism.

The paper aligns well with the purpose of the journal. However, there are some issues to be addressed in its current form:

  1. The authors mention that the silkworms were exposed to sublethal doses of avetmectin; however, in the introduction, no article is cited, nor is it mentioned how the doses were identified. 
  2. In the article, the impact of avermectin on chromatin structure is discussed, however, its impact on the silkworm phenotypic and economic parameters are not mentioned.
  3. A conclusion section should be integrated into the article and should comprise future perspectives (for instance, its applicability for other pesticides).

Author Response

Comments 1: The authors mention that the silkworms were exposed to sublethal doses of avetmectin; however, in the introduction, no article is cited, nor is it mentioned how the doses were identified.
Response 1: Thank you for pointing this out. We agree with this comment. Therefore, we have added detailed information on the sublethal dose and provided a brief explanation of why this dose was selected. The specific revisions are as follows:
We have added “Previous studies have indicated that the LC50 value of avermectin for 4-day-old larvae is 16.0 μg/L after a 96-hour exposure. A half-LC50 dose is commonly used as a sublethal concentration because it balances minimal mortality with significant physiological stress, enabling the study of early molecular and phenotypic responses without the confounding effects of high mortality.” to briefly explain why this dose was chosen on page 2, lines 81-85. The related reference was also added in the revised manuscript on page 12, lines 411-413.

Comments 2: In the article, the impact of avermectin on chromatin structure is discussed, however, its impact on the silkworm phenotypic and economic parameters are not mentioned.
Response 2: Thank you for pointing this out. We agree with this comment. Therefore, we have added the changes and characteristics observed in silkworms after exposure to avermectin. The specific revisions are as follows:
We have added “Based on the 96-hour median lethal dose (LC50) of 16 μg/L avermectin, an 8 μg/L concentration was selected for this experiment. After three days of feeding on the treated leaves, the silkworms exhibited severe toxic symptoms, including anorexia, salivation, body twisting into C or S shapes, and eventual immobility leading to death. In contrast, the control group, fed with untreated leaves, showed normal growth and development. These results indicate that even low concentrations of avermectin cause significant toxicity in silkworms.” on page 5-6, lines 190-199.

Comments 3: A conclusion section should be integrated into the article and should comprise future perspectives (for instance, its applicability for other pesticides).
Response 3: Thank you for pointing this out. We agree with this comment. Therefore, we have added a conclusion section “Our study shows how avermectin affects the 3D genome structure and gene expression in the silkworm midgut. Using Hi-C, ATAC-seq, and RNA-seq, we found changes in chromatin organization, increased accessibility, and shifts in genes related to detoxification and stress response. This study reveals a clear link between pesticide stress and genome architecture in silkworms. Future research could explore how other pesticides impact different non-target organisms. Comparing responses across species will help identify common and unique stress mechanisms. Studying long-term and inherited effects of pesticide exposure on genome structure is also important.” on page 10, lines 335-343.